# Subretinal mononuclear phagocytes induce cone segment loss via IL-1β

Chiara M Eandi[1,2,3,4†], Hugo Charles Messance[1,2†], Sébastien Augustin[1,2], Elisa Dominguez[1,2], Sophie Lavalette[1,2], Valérie Forster[1,2], Shulong Justin Hu[1,2], Lourdes Siquieros[1,2], Cheryl Mae Craft[5,6,7,8], José-Alain Sahel[1,2,9], Ramin Tadayoni[1,2,10], Michel Paques[1,2,9], Xavier Guillonneau[1,2‡], Florian Sennlaub[1,2*‡]

[1]Sorbonne Universités, UPMC University Paris 06, INSERM, CNRS, Paris, France; [2]Institut de la Vision, Paris, France; [3]Department of Clinical Science, University of Torino, Torino, Italy; [4]Eye Clinic, University of Torino, Torino, Italy; [5]Mary D. Allen Laboratory for Vision Research, Keck School of Medicine of the University of Southern California, Los Angeles, United States; [6]University of Southern California Eye Institute, Keck School of Medicine of the University of Southern California, Los Angeles, United States; [7]Department of Ophthalmology and Cell, Keck School of Medicine of the University of Southern California, Los Angeles, United States; [8]Department of Neurobiology, Keck School of Medicine of the University of Southern California, Los Angeles, United States; [9]Centre Hospitalier National d'Ophtalmologie des Quinze-Vingts, INSERM-DHOS CIC 503, Paris, France; [10]Department of Ophthalmology, Hôpital Lariboisièr, Paris, France

*For correspondence: florian. sennlaub@inserm.fr

†These authors also contributed equally to this work
‡These authors also contributed equally to this work

Competing interests: The authors declare that no competing interests exist.

**Abstract** Photo-transduction in cone segments (CS) is crucial for high acuity daytime vision. For ill-defined reasons, CS degenerate in retinitis pigmentosa (RP) and in the transitional zone (TZ) of atrophic zones (AZ), which characterize geographic atrophy (GA). Our experiments confirm the loss of cone segments (CS) in the TZ of patients with GA and show their association with subretinal CD14[+]mononuclear phagocyte (MP) infiltration that is also reported in RP. Using human and mouse MPs in vitro and inflammation-prone *Cx3cr1*[GFP/GFP] mice in vivo, we demonstrate that MP-derived IL-1β leads to severe CS degeneration. Our results strongly suggest that subretinal MP accumulation participates in the observed pathological photoreceptor changes in these diseases. Inhibiting subretinal MP accumulation or Il-1β might protect the CS and help preserve high acuity daytime vision in conditions characterized by subretinal inflammation, such as AMD and RP.

## Introduction

The macula consists of a small cone-dominated fovea, responsible for high acuity vision, surrounded by a rod-dominated parafovea and peripheral retina. Photo-transduction in cone segments (CS) are crucial for cone function and therefore high acuity and daytime vision. In geographic atrophy (GA), a late form of age-related Macular Degeneration (AMD), an extending atrophic zone (AZ) forms, characterized by the loss of the retinal pigment epithelium (RPE, a monolayer of cells with important functions in photoreceptor homeostasis) and degeneration of the photoreceptor cell layer (*Sarks, 1976*). The initial lesion in GA often develops parafoveally (*Sarks et al., 1988*) and slowly expands through the central retina and fovea, which leads to a severe drop in visual acuity. In the AZ, despite the absence of the RPE, residual cones survive, but they lack the cone segments (CS) (*Bird et al., 2014*). Similar findings are observed in disciform subretinal scars, the end-stage of

exudative AMD (*Curcio, 2001*). Surprisingly, in a transitional zone (TZ), just peripheral to the RPE-loss in the scar and AZ, the number of rods drops dramatically compared to the retina more distant from the lesion (*Bird et al., 2014*; *Curcio, 2001*). In contrast, the number of cones changes little in the TZ, but they lack their CS (*Bird et al., 2014*; *Curcio, 2001*). One could assume that the CS loss in the TZ of AMD patients is due to RPE dysfunction, but CS loss is also observed in retinitis pigmentosa (RP) patients with rod-gene mutations and unremarkable RPE (*Mitamura et al., 2013*).

Mononuclear phagocytes (MP) comprise a family of cells that include classical monocytes (Mo), macrophages (Mφ), and microglial cells (MC) among others (*Chow et al., 2011*). We and others have shown that subretinal MPs accumulate in the AZ and on the apical side of the RPE of the TZ in patients with GA (*Combadière et al., 2007*; *Gupta et al., 2003*; *Lad et al., 2015*; *Levy et al., 2015*). We have recently demonstrated that numerous blood-derived MPs (CCR2-positive) invariably participate in the infiltration of both, the AZ and TZ (*Sennlaub et al., 2013*). In mice, the recruitment of blood-derived monocytes contributes importantly to autoimmune, photo-oxidative, and genetic photoreceptor degeneration (*Rutar et al., 2012*; *Suzuki et al., 2012*; *Kohno et al., 2013*; *Cruz-Guilloty, 2013*) and we demonstrated that bone-marrow derived murine monocytes induce rod apoptosis in vivo and in vitro (*Sennlaub et al., 2013*), which is in part mediated by IL-1β (*Hu et al., 2015*). Interestingly, subretinal MPs also accumulate in RP secondarily to primary rod cell death, and have been suggested to induce the unexplained CS degeneration in these patients (*Gupta et al., 2003*).

We here confirm that the TZ is infiltrated by CD14[+]MPs and that rod photoreceptors and cone segments are lost in the TZ despite the presence of underlying RPE. Using co-cultures of human CD14[+]Mos and mouse bone-marrow derived Mos with retinal explants in vitro and inflammation-prone *Cx3cr1*[GFP/GFP] mice in vivo, we show that MP derived IL-1β induces CS degeneration additionally to previously reported rod apoptosis. Taken together, our results suggest that the presence of CD14[+]MPs in the subretinal space, not only observed in the TZ but also in patients with RP, participates in CS degeneration as they produce similar changes in vitro and in vivo.

## Results

### Cone and rod loss in relation to GA lesions

Bird and colleagues recently analyzed the loss of outer nuclear layer (ONL) nuclei and cone outer segments in GA using histology and electron microscopy (*Bird et al., 2014*). They reported that rod loss and cone outer segment degeneration occur peripheral to the margin of the RPE defect, in the transitional zone (TZ) (*Bird et al., 2014*). In this report, we analyzed the rod and cone populations in and around 10 atrophic zones (AZ) of 9 GA donor patients and in 6 control, non GA donor eyes using immunohistochemical techniques on paraffin sections with specific primary antibodies for rhodopsin, L/M-cone opsin, and cone arrestin, which allows identifying rod or cone photoreceptors even if they lost their outer segments. In the central region of control eyes, cone arrestin (green staining) -rhodopsin (red staining) double labeling visualized a 4 to 6 nuclei thick ONL, with cone segments (CS) and dendrites clearly demarcated by the cone arrestin immunological staining (*Figure 1A*, Hoechst blue staining of nuclei, RPE autofluorescence orange). The rod outer segments (OS) are strongly immunologically positive for rhodopsin staining, while the rod cell bodies stained more faintly, but enough to identify rod cell bodies (*Figure 1A* inset). The retina distant (>1000 μm) from the RPE lesion in sections from donor eyes with GA was very similar to the control eyes (*Figure 1B*), featuring clearly marked cone arrestin[+] CS (*Figure 1B* inset). In the TZ, close to the margin of the lesion but where RPE was still present, the ONL was irregular, thinned (around 2 nuclei of ONL), and cone and rod morphology was severely altered with the outer segments missing and the inner segments difficult to distinguish from the cell bodies (*Figure 1C*). The thinning and loss of segments was invariably found to extend peripherally to the margin of the RPE loss, into the TZ (*Figure 1C*, the autofluorescent RPE is visible in the left part of the micrograph, the arrow depicts the margin of the RPE with the atrophic zone to the right). The TZ, characterized by a thinned ONL in the presence of underlying RPE, was of variable length (200–800 μm) in our samples similar to previous reports (*Bird et al., 2014*). Within the AZ residual cone arrestin[+] cones, but also rhodopsin[+] rods were observed in all but one analyzed samples, but their distribution was irregular and all photoreceptors lacked their OS (*Figure 1D*). L/M-cone opsin immunohistochemistry, which recognizes the opsins of the most abundant red and green cones, confirmed the CS loss in the TZ and AZ

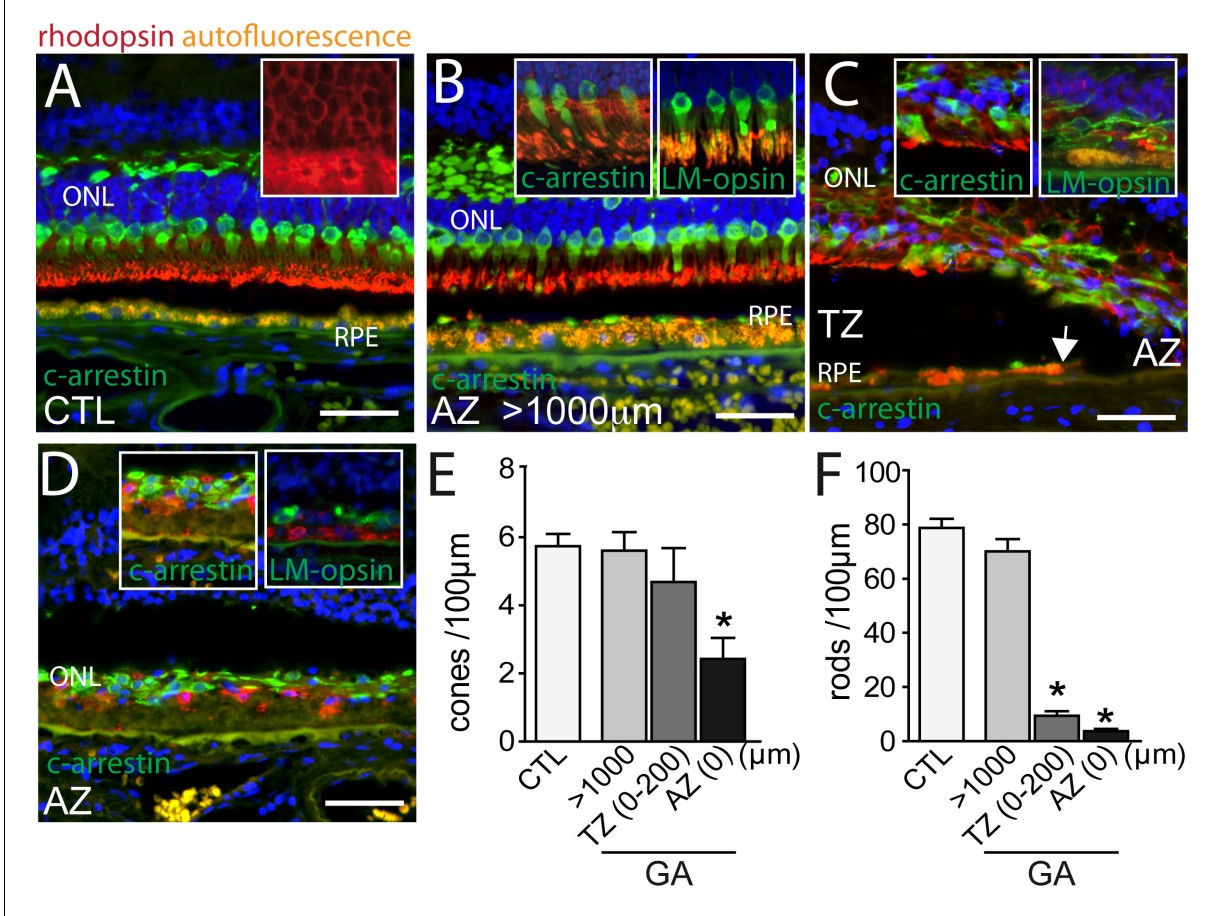

**Figure 1.** Rhodopsin, cone arrestin, and L/M cone opsin staining on central sections from control and geographic atrophy patients. (A–D) Representative micrographs of immunohistochemical detection of rhodopsin (red), cone arrestin (green, **A**–**D** and left insets of **B**, **C** and **D**), or L/M cone opsin (green, right insets of **B**, **C** and **D**) and Hoechst nuclear stain (blue, RPE autofluorescence visible in orange) on central sections of control donors (**A**) and donors with geographic atrophy (GA, panel **B**–**D**) at a distance greater than 1000 µm from the atrophic zone (AZ) (**B**), at the boundary of the AZ and transitional zone (TZ, panel **C**; white arrow indicates the margin of the autofluorescent RPE), and within the AZ (**D**). (**E** and **F**) Quantification of the number of arrestin[+]cone somata (**E**, one way ANOVA, Dunnett's post test *p=0,0024) and rhodopsin[+]rod somata (**F**, one way ANOVA, Dunnett's post test *p<0.0001) in control- and GA-donors (10 GA samples from 9 donors and in 6 control samples from 6 donors). ONL: outer nuclear layer; c-arrestin: cone-arrestin; RPE: retinal pigment epithelium; CTL: control; AZ: atrophic zone; TZ: transitional zone; GA: geographic atrophy. Scale bar = 50 µm.

(*Figure 1C,D* insets). Next, we quantified cone arrestin[+]cones (*Figure 1E*) and rhodopsin[+] rods (*Figure 1F*) in the central retina of control eyes and eyes with GA lesions. The cone density at a greater than 1000 µm distance from the RPE-denuded AZ, was similar between control and GA eyes. In the atrophic zone the number of cone somata was significantly reduced to around half the numbers of controls (*Figure 1E*) and was not significantly different in the TZ (0–200 µm from the AZ, which is the TZ length invariably found in all donor eyes). Rod cell counts, on the other hand, revealed a slight decrease distant from the lesions, but a severe 90% reduction in the TZ compared to controls (*Figure 1F*). Interestingly, in all but one atrophic zone we also detected rhodopsin[+] residual rods.

Taken together, our results confirm a severe rod cell loss in the TZ of patients with GA, where the RPE is still morphologically intact, while the number of cone numbers was relatively spared. However, cones of the TZ and residual rods and cones in the AZ had invariably lost their segments.

## Subretinal CD14[+] mononuclear phagocytes associate with cone segment loss in the transitional zone

Using pan- mononuclear phagocyte (MP) markers, we and others have previously shown that the subretinal space (i) above large Drusen (*Lad et al., 2015*; *Levy et al., 2015*; *Sennlaub et al., 2013*), (ii) within GA lesions (*Combadière et al., 2007*; *Gupta et al., 2003*; *Lad et al., 2015*; *Sennlaub et al., 2013*; *Penfold et al., 2001*), and (iii) in the TZ of patients with GA (*Levy et al., 2015*; *Sennlaub et al., 2013*) are infiltrated with MPs. We demonstrated that blood-born classical CD18[+]CCR2[+]Mos take part in the infiltrate at these three locations (*Sennlaub et al., 2013*). Circulating classical Mos also express high levels of CD14 (*Geissmann et al., 2003*), which is additionally expressed by other members of the MP family (*Gautier et al., 2012*) and is therefore not discriminative for infiltrating Mos versus resident MPs (unlike CCR2). We analyzed the presence of subretinal CD14[+]MPs and cone segment morphology on central RPE/choroid and retinal flat-mounts from 4 age-matched control donor eyes (*Figure 2A–D*) and 5 donor eyes with central GA lesions (*Figure 2E–H*), with particular attention to the TZ. Flat-mount immunohistochemistry facilitates the appreciation of cone outer segments and the detection of the smaller, dispersed MPs that are more difficult to detect on sections. We used CD14 immunohistochemistry to visualize MPs, peanut agglutinin (PNA) that stains inner and outer CS (but not cone cell bodies) (*Blanks and Johnson, 1984*),

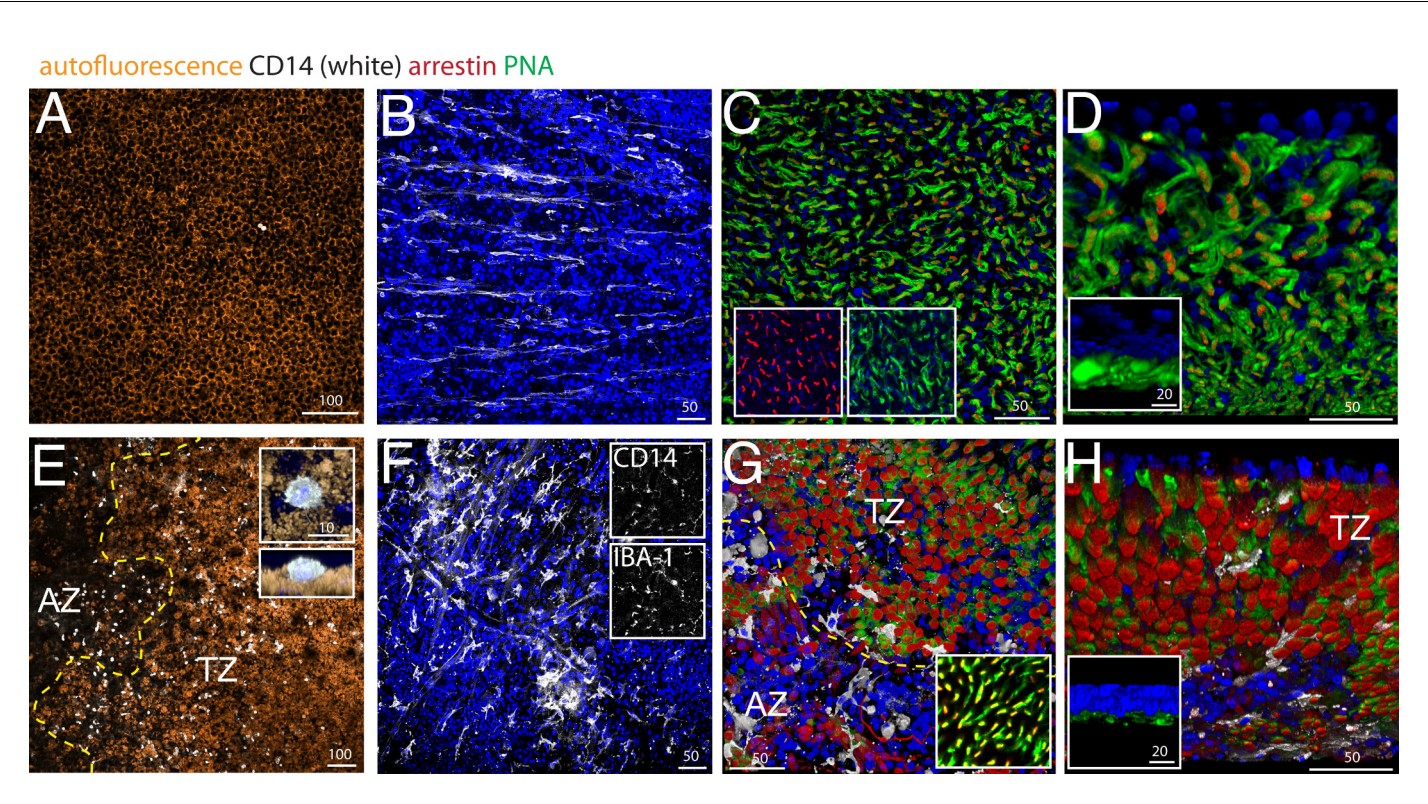

**Figure 2.** CD14, peanut agglutinin and cone arrestin staining on central flatmount preparations from control and geographic atrophy patients. Representative micrographs of immunohistochemical detection (confocal Z stack projections) of CD14 (white), cone arrestin (red), peanut agglutinin (PNA,green), and Hoechst nuclear stain (blue, RPE autofluorescence visible in orange) of RPE/choroid- (A and E) and retinal- flatmounts (B–D and F–H) of a control donor (A–D) and a donor with geographic atrophy (GA, panel E–H). (E) 3D reconstruction at higher magnification of a CD14[+]cell on RPE flatmounts in an orthogonal (E, upper inset) and perpendicular (E, lower inset). (F) Representative confocal micrographs of CD14 (F, upper inset) -IBA-1 (F, lower inset) double labeling. (D and H) Oblique and perpendicular (insets) 3D reconstruction views of the outer aspect of the retinal flatmounts. The margin of the atrophic zone (AZ), recognized by the loss of RPE (E) and by the irregular cone distribution and a thinned outer nuclear layer (G) is indicated by the yellow dotted line. (G) PNA/arrestin pattern distant (>1000 μm) from the AZ of a patient with GA (G inset). Experiments on flatmounts from 4 different control and patients with GA gave similar results. AZ : atrophic zone; TZ : transitional zone. Scale bars A and E = 100 μm; B–C and F–G = 50 μm; inset D and H = 20 μm.

and cone arrestin that stains cones irrespective of the presence of CS (see *Figure 1*). Central (1.5 cm around the fovea/center) retina/RPE/choroid complexes were dissected from the donor eyes and sectioned by radial incisions into 8 triangular pieces that each contain a parafoveal part in the case of control eyes and a central AZ, TZ, and non-atrophic more peripheral part for GA donor eyes. In GA tissues, the retinas of the AZ were carefully peeled from the RPE/choroid, as they adhere to Bruchs membrane in the area of RPE defects. This is not the case in the TZ, where the retina detaches easily from the underlying RPE, comparable to control retinas.

Confocal microscopy of CD14 stained central RPE confirmed that subretinal CD14$^+$MPs are only very occasionally observed in healthy age-matched donors (*Figure 2A*, CD14 white staining, RPE orange autofluorescence). Within the AZ of patients with GA where the RPE has disappeared (*Figure 2E* right to the yellow dotted line, AZ), CD14$^+$MPs were numerous, but were also always observed on the autofluorescent RPE in the TZ (*Figure 2E* left to the yellow dotted line, TZ), very similar to our published results using an IBA-1 antibody (*Levy et al., 2015*). A three dimensional reconstruction of an orthogonal and perpendicular close up view of a CD14$^+$cell shows its position on the apical side of the CD14-negative RPE (*Figure 2E* insets). The RPE also expresses CD14, but to a much lesser extend compared to MPs (*Elner et al., 2003*), which might explain why the RPE did not appear CD14-positive in our experimental conditions. A comparative RT-PCR of CD14 in human blood-derived Mos versus human post-mortem RPE cells revealed a ten-fold stronger expression in Mo in our samples (data not shown).

On retinal flatmounts, confocal microscopy of CD14 (white staining) / cone arrestin (red staining) / PNA (green staining) / Hoechst nuclear stain (blue staining) immunohistochemistry of control retina shows the CD14$^+$-resting MCs of the ganglion cell layer with their long processes (*Figure 2B*), while CD14$^+$MCs of the central inner retina of patients with GA displayed an activated MC phenotype with shortened processes (*Figure 2F*). CD14$^+$MCs also stained positive for the marker IBA-1, confirming their MP nature (*Figure 2F* insets). Orthogonal Z stack projections (*Figure 2C*), and oblique (*Figure 2D*) and perpendicular 3D reconstruction views (*Figure 2D* inset) of the outer aspect of the retinal flat-mounts reveal the normal cone arrestin$^+$PNA$^+$ segments of the central parafoveal retina in the absence of CD14$^+$MPs. Retinal flat-mounts of the AZ and adjacent TZ of patients with GA (*Figure 2G* below and above the yellow dotted line, respectively) show CD14$^+$MPs that stayed attached to the retina (white staining). In the AZ, recognizable by a severely thinned ONL and a slightly brown color in bright-field microscopy due to RPE-debris sticking to the subretinal aspect of the flat-mount, the PNA staining was much reduced, but cone arrestin$^+$cone cell bodies were visible in reduced numbers and their distribution was irregular (*Figure 2G*). In the adjacent TZ cone arrestin$^+$ cone density was relatively spared (*Figure 2G*). However, in both the AZ and TZ the typical cone arrestin$^+$PNA$^+$cone segment pattern had disappeared and cone arrestin immunological staining pattern of the remaining cone somata became apparent in the flat-mount, possibly because of greater antibody penetration in the thinned retinal flatmount. At a greater distance to the atrophic zone (>1000 μm), cone arrestin and PNA immunological staining located to cone segments, was indistinguishable to the pattern observed in flat-mounts from healthy donors (*Figure 2G* inset). Oblique (*Figure 2H*) and perpendicular 3D reconstruction views (*Figure 2H* inset) reveal the nearly complete loss of normal PNA$^+$ cone segments in the transitional zone that is associated with the presence of CD14$^+$MPs.

In summary, these results confirm the loss of cone inner and outer segments adjacent to the atrophic zone where the RPE is present that we observed in sections (*Figure 1*). They illustrate the close physical association of cone segment loss and CD14$^+$Mos accumulation in the TZ.

## Mononuclear phagocytes induce cone segment loss in retinal explants ex vivo

We have previously shown that wildtype- and in particular Cx3cr1-deficient-mouse bone marrow-derived monocytes (BMM) induce photoreceptor cell apoptosis in a monocyte/ mouse retinal explant co-culture system (*Sennlaub et al., 2013*). To analyze an eventual effect of human MPs on photoreceptors, we first co-cultured 100 000 human blood-derived CD14$^+$Mos (hMo) adhering to polycarbonate filters floating on DMEM with C57BL/6 mouse retinal explants for 18 hr (with the photoreceptors facing the adherent Mos). Apoptosis was analyzed by TUNEL staining of the retinal explants cultured without Mos (*Figure 3A*), or with Mos (*Figure 3B*). TUNEL$^+$ nuclei in the photoreceptor cell layer of retinal explants were more numerous in the presence of Mos. Double-

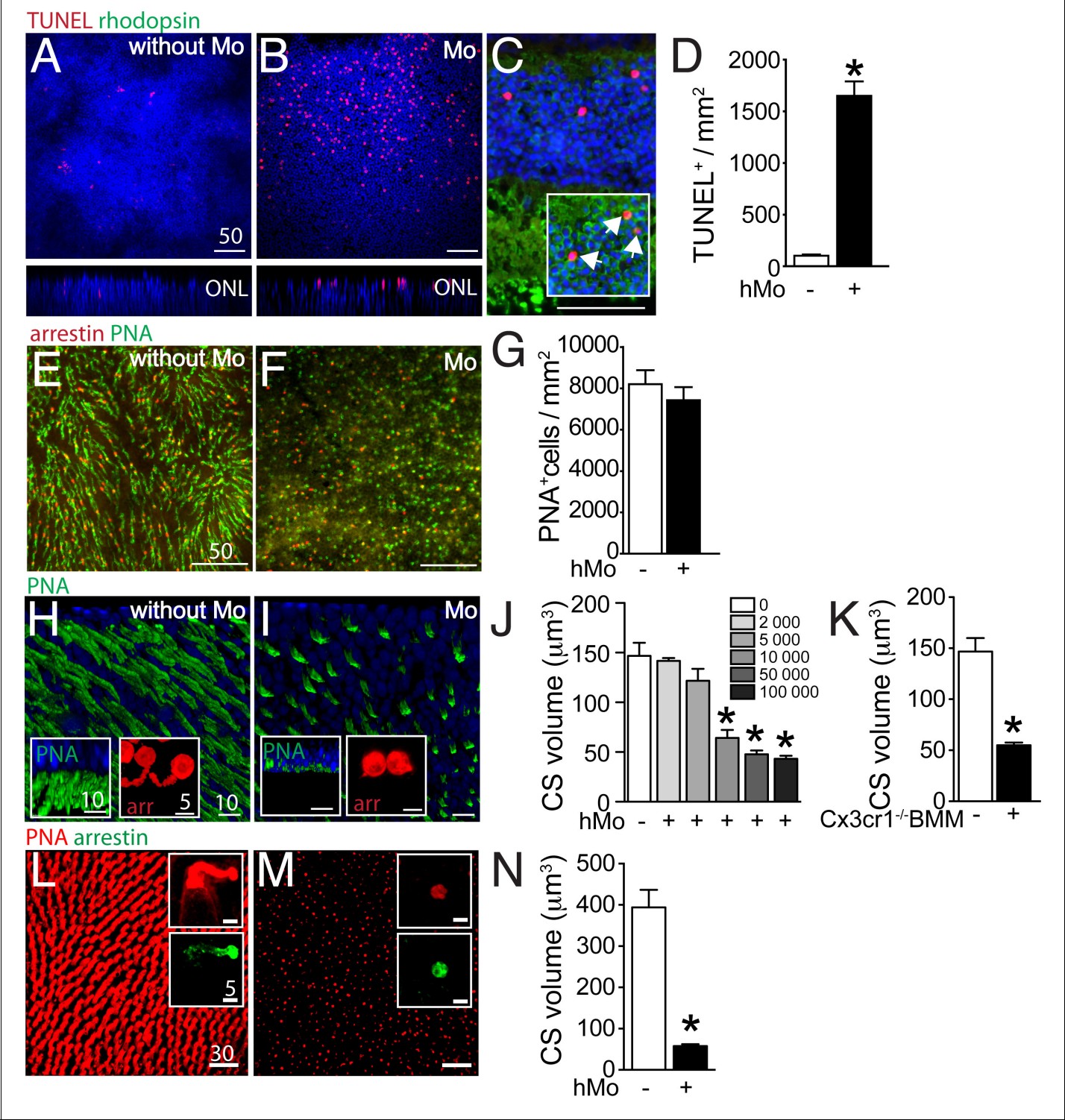

**Figure 3.** Evaluation of rods and cones in monocyte/retinal explant co-cultures. (A and B) Orthogonal projection of confocal Z stack images of the outer nuclear layer of TUNEL- (red), Hoechst nuclear dye- (blue) stained mouse retinal explants after 18 hr of culture (A) or co-culture with human monocytes (hMo, panel B). (C) TUNEL(red)/rhodopsin(green) co-staining of a section of a mouse retinal explant co-cultured with hMo (the inset represents a longer green exposure of the outer nuclear layer). (D) Quantification of TUNEL+ nuclei in mouse retinal explants cultured with or without hMo (n = 4/group, Mann Whitney *p=0,028). (E and F) Orthogonal projection of confocal Z stack images of the photoreceptor segments of mouse retinal explants after 18 hr of mono-culture (E) or co-culture with hMo (F) after cone arrestin (red)/peanut agglutinin (PNA, green) staining. (G) Quantification of cone numbers in retinal explants cultured with or without hMo (n = 10/group). (H and I) Oblique-, and perpendicular (insets) -3D
*Figure 3 continued on next page*

*Figure 3 continued*

reconstruction views of confocal Z stack images of the photoreceptor segments of mouse retinal explants after 18 hr of mono-culture (**H**) or co-culture with hMo (**I**) after peanut agglutinin (PNA, green) and cone-arrestin (red, insets) staining. (**J**) Quantification of cone segment volume in mouse retinal explants cultured without or with the indicated numbers of hMos (n = 6/group, Mann Whitney *p<0,0004). (**K**) Quantification of cone segment volume in mouse retinal explants cultured without or with 100 000 Cx3cr1-deficient bone marrow-derived Mos (n = 4–6/group, Mann Whitney *p<0,0061). (**L** and **M**) Orthogonal-, and perpendicular (insets) -3D reconstruction views of confocal Z stack images of the photoreceptor segments of macaque retinal explants after 18 hr of mono-culture (**L**) or co-culture with hMo (**M**) after peanut agglutinin (PNA, red) and cone-arrestin (green) staining. (**N**) Quantification of cone segment volume in macaque retinal explants cultured without or with 100 000 hMos (n = 8/group, Mann Whitney *p<0,0002). hMo/Mo : human monocyte; arr: cone-arrestin; CS : cone segments; PNA : peanut agglutinin. Scale bar : **A–F** = 50 µm; **H** and **I** = 10 µm.

immunological labeling for rhodopsin and TUNEL on sections prepared from the retinal explants revealed that the TUNEL$^+$ nuclei of the ONL were positive for rhodopsin (*Figure 3C* inset) and not located at proximity to the inner segments, where cone nuclei are located (*Figure 3C*). Apoptosis was only observed in a minority of rods in these short-term co-cultures and rhodopsin$^+$rod segments were unremarkable in the culture condition (*Figure 3C*). Quantification of TUNEL$^+$ nuclei/mm on 8 explants per group revealed a significant increase in TUNEL$^+$ photoreceptors in co-culture with hMos compared to retinal explants cultured without hMos (*Figure 3D*) similar to co-cultures using Cx3cr1-deficient BMMs (*Sennlaub et al., 2013*). Next, we analyzed the cone population of PNA (green fluorescence) cone arrestin (red fluorescence) double-labeled mouse retinal explants after 18 hr of mono-culture (*Figure 3E*), or co-culture with hMos (*Figure 3F*). Quantification of cone arrestin$^+$PNA$^+$ cones (*Figure 3G*) revealed no influence of the presence of hMos on the number of cones after 18 hr of culture. However, the PNA-staining in hMo-exposed retinal flat-mounts appeared punctuated and the typical elongated shape of mouse PNA$^+$CS staining in retinas cultured without Mos had disappeared (*Figure 3E and F*). Oblique and perpendicular 3D reconstruction views of confocal microscopy Z-stacks of explant mono-cultures (*Figure 3H*), or co-culture with hMos (*Figure 3I*) reveals the near complete loss of PNA$^+$CS in the hMo exposed explants. Similarly, arrestin staining revealed a severe shortening of arrestin$^+$CS in the co-culture condition compared to controls (*Figure 3H and I* insets). Quantification of PNA$^+$ volume (quantitated on the whole Z-stack, divided by the number of cones) in retinal explants co-cultured with increasing numbers of hMo confirms numerically a severe, significant reduction in PNA$^+$CS volume as a function of hMos numbers (*Figure 3J*). To evaluate eventual artifacts due to interspecies incompatibilities we next quantified PNA$^+$CS volumes of mouse retinal explants that we co-cultured for 18 hr with 100 000 Cx3cr1-deficient-mouse BMMs (which induces rod apoptosis similar to hMos [*Sennlaub et al., 2013*]). Our results revealed a similar, severe reduction in CS volume (*Figure 3K*). Furthermore, orthogonal 3D reconstruction views of confocal microscopy Z-stacks of PNA(red)/arrestin(green)-stained explants prepared from the para-central area of a non-human primate cultured without (*Figure 3L*), or with 100 000 hMos (*Figure 3M*) reveals a similar near complete loss of PNA$^+$CS in the hMo exposed primate explants (Quantification *Figure 3N*). Again, the arrestin staining revealed a similar severe shortening of arrestin$^+$CS (*Figure 3L and M* insets).

Taken together, our data show that Mos induce rod apoptosis and severe CS loss in a short-term retinal explant model. The CS loss was observed with the PNA and arrestin staining, with human blood-derived Mo and Cx3cr1-deficient-mouse BMMs, on mouse retinal explants and in explants from non-human primates, suggesting that the cone alterations were not the result of interspecies incompatilibilities.

## IL-1β induces cone segment loss in retinal explant

Mos were not observed to physically infiltrate the outer segments or to ingest rhodopsin$^+$ or PNA$^+$ segments in the 18 hr co-culture (data not shown). Alternatively, the rapid CS reduction in the co-culture system might be due to the production of a secreted factor, such as inflammatory cytokines. We have previously shown that Cx3cr1-deficient mouse BMM induce rod apoptosis in the Mo/retinal explant co-cultures via IL-1β (*Hu et al., 2015*). We showed that Cx3cr1-deficient BMMs constitutively secrete adenosine triphosphate (ATP), express increased surface P2RX7 receptor and secrete mature IL-1β after transcriptional stimulation without a second exogenous stimulus (*Hu et al., 2015*), that is classically required for inflammasome activation and IL-1β maturation (*Schroder and Tschopp,*

*2010*). Similarly, human blood monocytes constitutively release endogenous ATP, which leads to mature IL-1β after transcriptional induction without a second exogenous stimulus (*Netea et al., 2009*). RT-PCRs of mRNA prepared from freshly isolated hMo, hMo cultured alone or with a retinal explant for 18 hr (both on polycarbonate filters) revealed that IL-1β mRNA, but not IL18 mRNA, was significantly induced in hMos in the co-culture condition (*Figure 4A*), contrary to IL-6 and TNFα that were not induced (data not shown). PNA-staining of retinal explants cultured for 18 hr in the absence of Mos (*Figure 4B*) but with recombinant IL-1β (*Figure 4C*), showed that IL-1β was sufficient to severely reduce PNA⁺CS. Quantification of PNA⁺CS confirmed the significant, IL-1β-induced volume loss (*Figure 4D*). In these experiments IL-1β was used at a concentration of 50 ng/ml (in the range of concentration classically used in vitro) but the local IL-1β concentration that cones are exposed to in the co-culture is difficult to estimate, as a gradient of IL-1β is expected to form around the IL-1β producing Mos. To directly evaluate the effect of locally produced IL-1β we studied the effect of the IL-1 receptor antagonist (IL-1Ra) on the co-cultures. PNA-stained flatmounts of Mo/retinal co-cultures without (*Figure 4E*) or in the presence of IL-1Ra (*Figure 4F*) shows that IL-1β-inhibition significantly protects against Mo-induced CS loss (*Figure 4G*).

In summary, these experiments show that IL-1β, transcribed by Mos in the co-culture, is sufficient to induce CS reduction and necessary in Mo-induced CS loss.

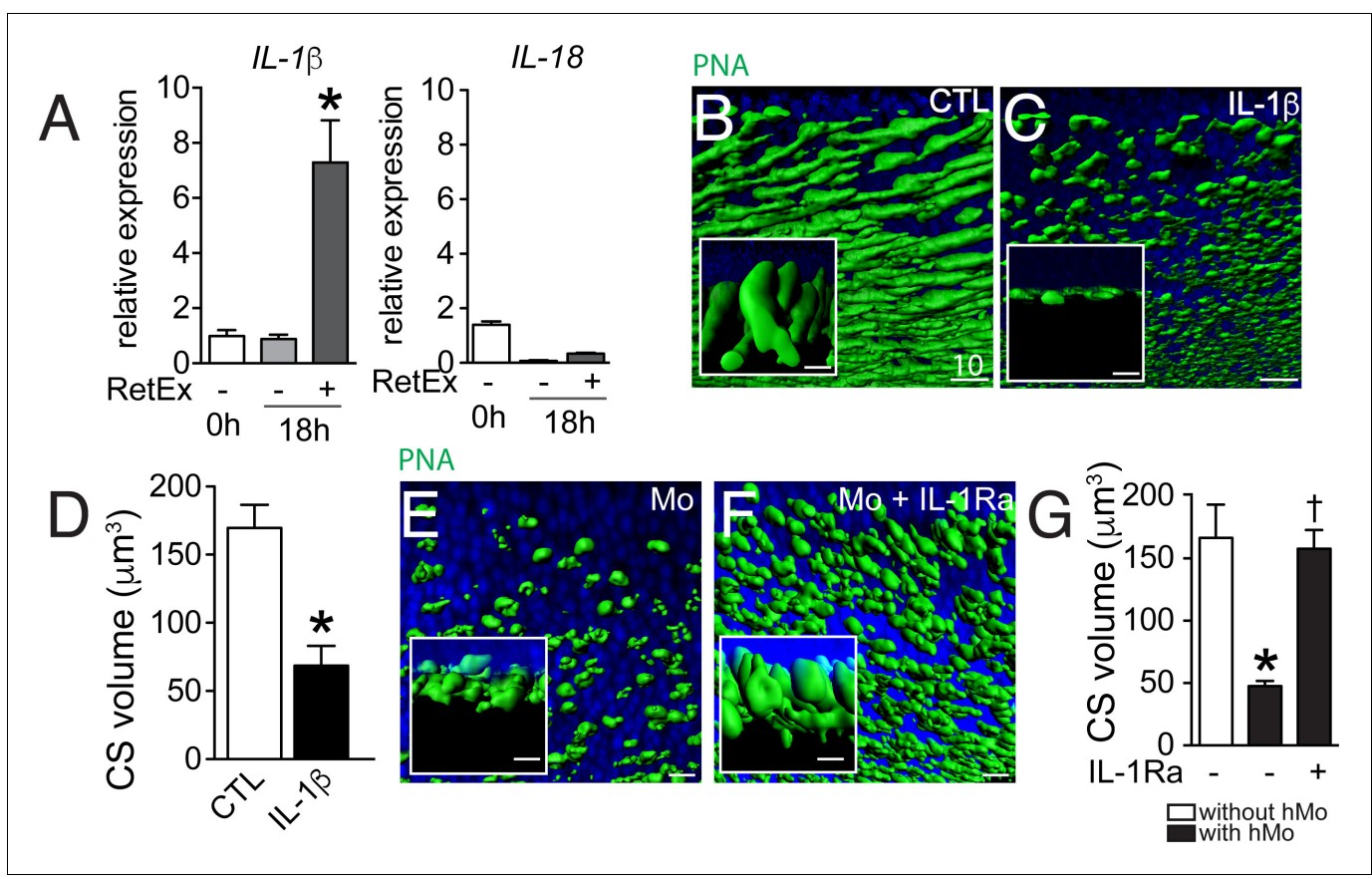

**Figure 4.** The influence of IL-1β retinal explants co-cultures. (A) Quantitative RT-PCR of RT-PCR of IL-1β and IL-18 normalized with *S26* mRNA in fresh human monocytes (hMo), hMo cultured alone or with a retinal explant for 18 hr (both on polycarbonate filters, n = 5, ANOVA, Dunnett's post test*p=0,0079). (B and C) Oblique and perpendicular (insets) 3D reconstruction views of 18 hr peanut agglutinin (PNA)-stained retinal explant (B) and IL-1β (50 ng/ml) exposed explant (C). (D) Quantification of cone segment volume in retinal explants cultured with or without IL-1β (n = 6/group, Mann Whitney *p=0,0087). (E and F) Oblique and perpendicular (insets) 3D reconstruction views of 18 hr peanut agglutinin (PNA)-stained retinal explant co-cultured with human monocytes without (E) or with IL-1 receptor antagonist (F, 10 mg/ml). (G) Quantification of cone segment volume in Mo/retinal co-cultures with or without an IL-1 receptor antagonist (n = 6/group, Kruskal-Wallis, Dunn's post test *p=0,0022 versus 'without hMo'; †p=0,0182 versus "with hMo without IL1-Ra). RetEx: retinal explant; CTL: control; CS: cone segment; hMo: human Monocytes. Scale bar B, C, E, F = 10 µm.

## IL-1Ra inhibits cone segment loss in subretinal inflammation in vivo

Our data show that CS loss in the TZ is associated with CD14[+]MP infiltration of the subretinal space in patients with GA and that Mos, and more precisely Mo-derived IL-1β, induce a similar effect in vitro. However, the in vitro co-culture system is by nature a very artificial model, characterized by an excess of Mos, the absence of an underlying RPE, and rod and cone segment degeneration occurring in hours rather than days and weeks. To evaluate if subretinal inflammation and IL-1β secretion observedin vivo induce CS loss in the presence of the RPE, we used a light-challenge model of *Cx3cr1*[GFP/GFP] mice. We have previously shown that *Cx3cr1* deficiency in mice (lacking the tonic inhibitory signal from CX3CL1 expressing neurons [*Zieger et al., 2014*; *Silverman et al., 2003*]) leads to a strong increase of subretinal MP accumulation with age from the age of 6 months (*Combadière et al., 2007*) or after light-challenge or laser-injury (*Combadière et al., 2007*; *Raoul et al., 2008*; *Ma et al., 2009*). We showed that the subretinal infiltrate in these mice is composed of a mixture of microglial cells and pathogenic blood-derived Mos similar to GA lesions (*Combadière et al., 2013*). We demonstrated that IL-1β protein is increased in *Cx3cr1*[GFP/GFP] mice with age (*Lavalette et al., 2011*) and after a 4 day acute light-challenge that it induces rod cell apoptosis without affecting RPE morphology (*Hu et al., 2015*). *Cx3cr1*[GFP/GFP] mice do not develop Drusen and RPE atrophy, but they do model MP accumulation between the RPE and the photoreceptor segments, as well as some degree of associated rod degeneration (*Combadière et al., 2007*; *Sennlaub et al., 2013*; *Hu et al., 2015*), somewhat similar to the TZ of patients with GA (*Figures 1* and *2*). Eventual cone alterations in *Cx3cr1*[GFP/GFP] mice have not yet been described. Orthogonal 3D reconstruction views of confocal microscopy Z-stacks of CS in PNA-stained retinal flat-mounts of 2–3 months old *Cx3cr1*[GFP/GFP] mice kept under normal light conditions (*Hu et al., 2015*) (*Figure 5A*) displayed the typical elongated shape of PNA[+]CS staining similar to freshly extracted wildtype retinal flat-mounts (data not shown). In contrast, PNA[+]CS of 4 day light-challenged *Cx3cr1*[GFP/GFP] mice that accumulate GFP[+] subretinal MPs, were shortened and punctuated (*Figure 5B*) throughout the retinal flatmounts. Daily subcutaneous IL-1Ra injections during the light-challenge of *Cx3cr1*[GFP/GFP] mice prevented the change in CS morphology despite the presence of GFP[+]MPs in the subretinal space (*Figure 5C*). Quantification of PNA[+]CS volume in control and light-challenged C57BL6/J and *Cx3cr1*[GFP/GFP] mice revealed a significant decrease in the light-challenged *Cx3cr1*[GFP/GFP] mice (*Figure 5D*). Strikingly, IL-1β inhibition, which had no effect on subretinal MP accumulation (*Figure 5E*), completely prevented the loss of PNA[+]CS volume in the light-challenged mice (*Figure 5D*).

Taken together, our data show that subretinal inflammation, observed in light-challenged *Cx3cr1*[GFP/GFP] mice in vivo is associated with IL-1β-dependent PNA[+]CS degeneration in the presence of the RPE similar to the results we obtained with Mos and recombinant IL-1β in vitro.

## Discussion

Our immunohistological study of human donor eyes reveals severe rod cell loss, little cone loss but CS degeneration peripheral to the margin of the RPE lesion that defines GA, confirming a recent morphological study (*Bird et al., 2014*). As previously reported, this TZ, characterized by a thinned outer nuclear layer despite the presence of the RPE, was of variable length but found in all specimens. We also confirm a surprisingly important number of residual cones that lack their CS in the AZ (*Bird et al., 2014*; *Curcio, 2001*). Interestingly, rhodopsin immunohistochemistry also identified some residual, segment-lacking rhodopsin[+] rods within the RPE lesion.

Other investigators and we have previously shown that AZs of patients with GA contain numerous subretinal MPs (*Combadière et al., 2007*; *Gupta et al., 2003*; *Lad et al., 2015*; *Sennlaub et al., 2013*; *Penfold et al., 2001*). We also demonstrated that IBA1[+]MPs, but also blood-derived CCR2[+]-MPs, invade the subretinal space of the TZ (*Levy et al., 2015*; *Sennlaub et al., 2013*). In the current study, we used immunohistochemistry on flat-mount preparations with an antibody recognizing a co-receptor of the toll like receptor 2 and 4, CD14, to visualize the infiltration of CD14[+]MPs (blood- or tissue-derived MPs) beyond the AZ in the TZ. Co-staining for PNA and cone arrestin confirmed the presence of residual cones in the AZ and the severe CS degeneration in the TZ of retinal flat-mounts and illustrates the close physical association of CS degeneration with CD14[+]MPs.

It is not clear why CS degenerate in the TZ, where the RPE is present. The degeneration might be a result of dysfunctional RPE prior to its degeneration, suggested by the observation of

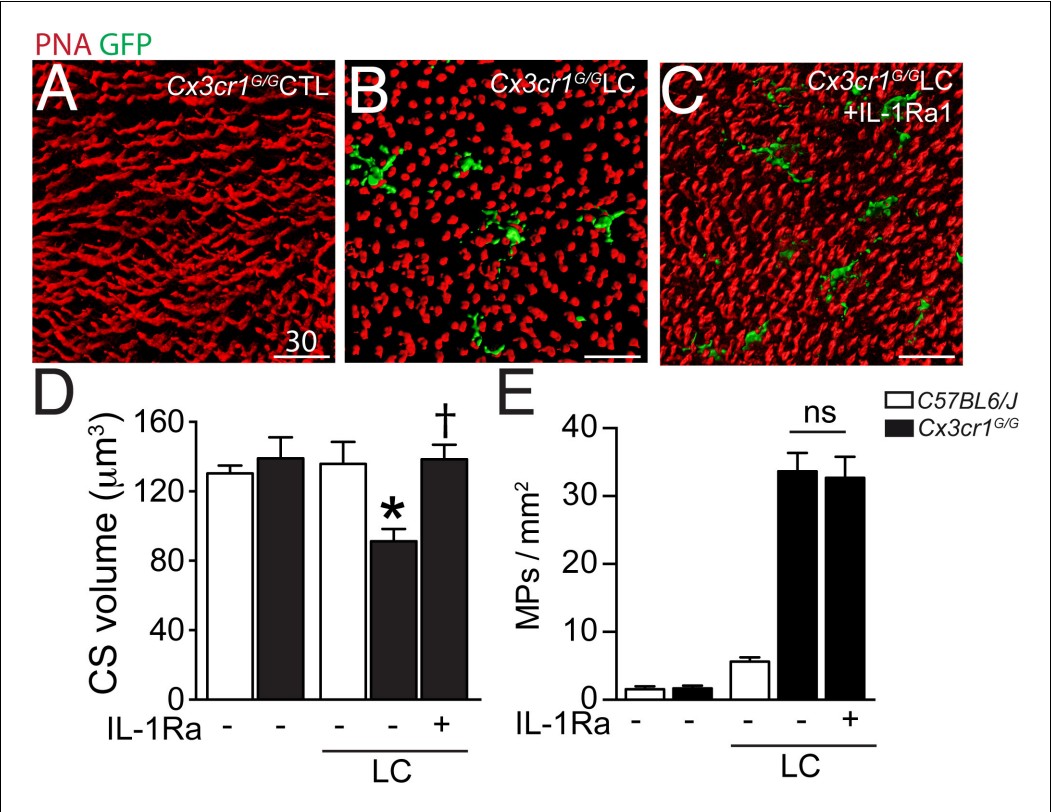

**Figure 5.** Cone segments in light-induced subretinal inflammation of C57BL6/J and *Cx3cr1GFP/GFP*-mice and the effect of pharmacological IL-1β inhibition. (**A–C**) Orthogonal projections of confocal Z stack images of the photoreceptor segments of peanut agglutinin (PNA, red) stained *Cx3cr1GFP/GFP*-mice (expressing GFP under the *Cx3cr1* promoter) that were kept under normal light conditions (**A**), or exposed to an inflammation-inducing light-challenge (LC) and treated with PBS (**B**) or IL-1β receptor antagonist (**C**, IL-1Ra). (**D** and **E**) Quantification of cone segment volume (**D**) and subretinal mononuclear phagocytes (**E**) in normal light raised and LC-mice (n = 5–8/group; Kruskal-Wallis, Dunn's post test *p=0,0109 versus *Cx3cr1GFP/GFP*-mice in normal light; †p=0,0028 versus light-challenged *Cx3cr1GFP/GFP*-mice). PNA: peanut agglutinin; GFP: green fluorescent protein; IL-Ra: IL-1β receptor antagonist LC: illuminated/light-challenged. Scale bar 30 μm.

hyperfluorescence of the TZ (*Sparrow et al., 2010*). However, to date, we do not know if the hyper-fluorescence is due to an increase of fluorescent bisretinoids in the RPE (which could be a sign of RPE dysfunction), or to the superposition of several RPE cells or RPE-debris containing subretinal MPs in the TZ (*Lad et al., 2015*; *Sennlaub et al., 2013*; *Gocho et al., 2013*; *Zanzottera et al., 2015*). The observation that CS degeneration is also observed in RP patients with rod-gene mutations and unremarkable RPE (*Mitamura et al., 2013*) further argues that an RPE-independent factor might participate in the degeneration.

Intriguingly, CS degeneration in RP (*Gupta et al., 2003*), and as we show here in the TZ of patients with GA , is associated with the accumulation of subretinal MPs. We have previously shown that mouse MPs induce partial rod cell apoptosis in vivo and that wildtype mouse Mos and Cx3cr1-deficient Mos and MCs (characterized by excessive IL-1β secretion) induce rod apoptosis in retinal explant co-cultures (*Sennlaub et al., 2013*; *Hu et al., 2015*). The influence of subretinal MPs on cones is unknown. Using Mo/retinal explant co-cultures, we here show that human blood-derived CD14[+]Mo induce some degree of rod apoptosis in retinal explants similar to mouse Cx3cr1-deficient mouse BMMs (*Hu et al., 2015*). Interestingly, cone death was not observed, but all cones showed signs of severe CS degeneration. We show that IL-1β was robustly transcribed in co-cultured Mos, that recombinant IL-1β replicated CS degeneration, and that IL-1β-inhibition reversed the Mo-induced CS degeneration.

The co-culture in vitro system, characterized by an excess of Mos, the lack of underlying RPE, mild rod apoptosis and quite severe CS degeneration occurring in hours rather than days and weeks, can however only be a rough approximation of the in vivo situation. The pattern of continuous rod (but not cone) apoptosis and CS degeneration over longer time periods would result in severe rod cell loss (as they don't regenerate), little cone loss but CS degeneration intriguingly similar to what is observed in the TZ of patients with GA.

To test whether IL-1β-induced CS degeneration takes place in subretinal inflammation in vivo in the presence of the RPE, we analyzed cone morphology in inflammation-prone $Cx3cr1^{GFP/GFP}$ mice. When subretinal MP accumulation was induced using a light-challenge (Hu et al., 2015), we observed CS degeneration, similar to the results we obtained with Mos in vitro Furthermore, inhibition of IL-1β, induced in these experimental conditions (Hu et al., 2015), prevented the CS degeneration in vivo, in accordance with the observation that IL-1β potently induced CS degeneration in vitro.

Our study clearly demonstrates the association of subretinal MPs with CS degeneration in human samples and that MPs can induce a similar phenotype in vitro and in vivo, which strongly suggest that subretinal MPs participate in the degenerative changes in GA. Further studies of IL-1β expression in human GA samples are needed to implicate this specific cytokine in the degenerative process in the human disease. However, IL-1β has been shown to be strongly expressed in subretinal MPs of choroidal neovascular membranes in wet AMD (Oh et al., 1999) and CS degeneration is also observed adjacent to the fibrovascular membranes in disciform subretinal scars of wet AMD (Curcio, 2001). These observations suggest that an IL-1β-dependent mechanism might exist in the human pathology.

Little is known about the cellular and molecular mechanisms of CS loss in GA. The observation that the degenerative changes appear peripheral to the RPE lesion and that similar changes are observed in RP (Gupta et al., 2003) and RP models (Punzo et al., 2009), suggest that other factors than RPE loss play a significant role. We clearly demonstrate that subretinal MPs, observed in both RP and the TZ, induce severe IL-1β-dependent CS degeneration in the presence of morphologically normal RPE in vivo. Gupta et al. proposed several years ago that the subretinal inflammation secondary to rod degeneration in RP could be the 'missing link' to explain secondary CS and cone degeneration observed in RP (Gupta et al., 2003). CS maintenance in RP has recently been shown to depend on cone glucose uptake and aerobic glycolysis (Aït-Ali et al., 2015) that is stimulated by insulin (Punzo et al., 2009) and rod-derived cone viability factor (Aït-Ali et al., 2015). Inflammatory cytokines, including IL-1β, inhibit insulin-dependent glucose uptake and are increasingly recognized to play an important role in insulin-resistance in the pathogenesis of type 2 diabetes (Shoelson, 2006). IL-1β-induced cone insulin resistance, decreased glucose uptake and subsequent cone starvation might explain the CS loss in RP and AMD.

Taken together, these results strongly suggest that infiltrating MPs observed in the TZ of patients with GA and in RP patients contribute importantly in the CS degeneration as they produce similar changes in vitro and in vivo. Inhibiting subretinal MP accumulation, or as we show IL-1β inhibition, might help preserve CS and high acuity daytime vision in these patients.

## Materials and methods

### Donor eyes and immunohistochemistry

Donor eyes with a known history of AMD and controls were collected through the Minnesota Lions Eye Bank: 9 control maculae from 9 patients (5 men and 4 women; mean age 85 years ± 2.4 SEM) and 15 GA donor maculae from 12 patients (5 men and 7 women; mean age 82.9 years ± 2.2 SEM). Donor eyes without a history of eye disease and with a visible fovea on post-mortem funduscopy and a regular histology of the parafovea, recognizable by multilayered ganglion cell nuclei were used as control eyes. GA donor eyes had a known history of AMD, visible GA lesions on post-mortem fundus pictures, and histologically confirmed missing RPE and thinned photoreceptor cell layer in the lesion of the parafovea (multilayered ganglion cell layer) in the absence of subretinal vessels (as observed in neovascular AMD) or a fibroglial scar. Donor families gave informed consent in accordance with the eye bank's ethics committee. Postmortem fundus photographs were taken and the posterior segment was fixed 4 hr in 4% PFA, dissected, embedded in paraffin, and sectioned. The

anterior segment was dissected at the Minnesota Eye Bank, fundus photographs were taken, the posterior segment was fixed in PFA 4% for 4 hr and placed in ice cold PBS for shipping to our laboratory.

For immunohistochemistry on paraffin sections (5 control maculae from 5 patients and 10 GA donor maculae from 7 patients), the posterior segments were dissected to include the optic nerve, the macula and the central retina between the vessel arches, included in paraffin, and sectioned. The sections of the retina from donor eyes were stained as follow. After rehydration, antigen retrieval was performed in 70% formic acid for 15 min. After a pre-incubation step in 10% horse serum diluted in PBS 1X, slides were incubated with primary antibodies (anti-rhodopsin antibody [clone 4D2, Chemicon-Merck Millipore, Guyancourt, France; 1:500]; rabbit polyclonal anti-human cone arrestin antibody [DEI Luminaire Founder, Cheryl Craft; 1:25000] (*Zhang et al., 2001*), L/M-cone opsin [Chemicon; 1:500]) and revealed with the appropriate secondary antibodies (Life Technologies, Villebon-sur-Yvette, France). Pictures of the retina were taken with a fluorescence microscope (Leica DM550B, Leica, Nanterre, France) and rods and cones quantified within the GA lesion, 0–200 µm and >200 µm from the lesion.

For flat-mount immunohistochemistry (4 age-matched control donor eyes and 5 donor eyes with central GA lesions), central (1.5 cm around the fovea/center) retina/RPE/choroid complexes were dissected from the donor eyes and sectioned by radial incisions into 8 triangular pieces that each contain a parafoveal part in the case of control eyes and AZ central, adjacent TZ, and non-atrophic peripheral part for GA donor eyes. In GA tissues, the retina of the atrophic part was subsequently carefully detached from underlying Bruch's membrane to which the atrophic retina is attached. Immunohistochemistry was performed on submerged samples with the primary antibodies anti-CD14 (MCA1218; 1:100 Serotec, Kidlington, UK), Peanut agglutinin Alexa 488 (Thermofisher, Villebon-sur-Yvette, France; 1:50), anti-human cone arrestin antibody (LUMIF-hCAR; 1:10000), Hoechst nuclear stain (Thermofischer; 1:1000), and revealed using appropriate fluorescent secondary antibodies (Molecular Probe). Preparations were observed with a Olympus FV1000 (Olympus, Rungis, France) confocal microscope.

## Monocyte preparations, monocyte-retinal co-culture, and retinal explants

PBMCs were isolated from heparinized venous blood from healthy volunteer individuals. In accordance with the Declaration of Helsinki, volunteers provided written and informed consent for the human monocyte expression studies, which were approved by the Centre national d'ophthalmologie des Quinze-Vingt hospital (Paris, France) ethics committees (no. 913572). Peripheral blood mononuclear cells (PBMCs) were isolated from blood by 1-step centrifugation on a Ficoll Paque layer (GE Healthcare) and sorted with EasySep Human Monocyte Enrichment Cocktail (StemCells Technology, Grenoble, France). Human Mo were seeded on polycarbonate filters floating on DMEM for 2 hr. *C57BL/6J* mouse retina were prepared and placed with the photoreceptors facing 2000 to 100,000 adherent Mo for 18 hr at 37°C. In one set of experiments IL-1β was inhibited by the IL-1 receptor antagonist (R&D, Lille, France, 10 mg/ml). Alternatively, retinal explants were cultured with recombinant IL-1β (R&D; 50 ng/ml). In one set of experiments, we exposed eight 5 × 5 mm (*Sarks et al., 1988*) retinal explants dissected from the peri-central retina of a cynomolgus macaque to 100,000 human Mo or cultured eight explants without Mos. These experiments were done in accordance with the National Institutes of Health Guide for Care and Use of Laboratory Animals. The donor retinal tissue samples were prepared from a control eye of an unrelated protocol that was approved by the Local Animal Ethics Committees and conducted in accordance with Directive 2010/63/EU of the European Parliament and French authorization C 92-032-02 regulation. The non human primate used in this study were cynomolgus macaques (macaca fasicularis) from foreign origin. Macaca fasicularis were housed under standard environmental conditions (12-h light–dark cycle, 22 ± 1°C, 50% humidity) with free access to food and water. The animal was sacrificed by an overdose of barbiturate pentobarbital, the eyes were enucleated and retina dissected and prepared for retinal explant culture. For TUNEL staining (In Situ Cell Death Detection Kit, Roche Diagnostics, Meylan, France) mouse retinal flat-mounts were fixed in 4% PFA for 30 min, washed in 1x PBS (pH 7.3), incubated for 90 min at 37°C with the reaction mixture and the reaction was stopped by washing with 1x PBS. For immunohistochemistry the flat-mounts were incubated with anti-mouse cone-arrestin antibody (Millipore, Guyancourt, France, #AB15282, 1:100; mouse retina explants) or rabbit polyclonal anti-human cone

arrestin antibody (DEI Luminaire Founder, Cheryl Craft; 1:25000; macaque retinal explants) and Peanut agglutinin Alexa 488 (Thermofisher; 1:50; both mouse and macaque explants) overnight and revealed with an appropriate antibody (Thermofisher). Nuclei were stained with Hoechst (Thermofischer; 1:1000). Flat-mounts images were captured with a DM 5500 microscope (Leica) or an Olympus Confocal microscope. PNA$^+$cone S (CS) volume was quantified on confocal microscopy Z-stacks that encompass the 2 outermost nuclei of the ONL and the whole outer segments, and divided by the number of cones in the field.

## Reverse transcription and real-time polymerase chain reaction (RT-PCR)

RT-PCR was used to measure mRNA expression levels of *IL-1β*. Total RNA was extracted from fresh monocytes and monocytes cultured for 18 hr on polycarbonate filters floating on DMEM with or without an overlaying retinal explant using the Nucleospin RNA XS (740902, Macherey-Nagel, Düren, Germany) according to the manufacturer's instructions and converted to cDNA using oligo (dT) as primer and Superscript II (Life Technologies). Each RT assay was performed in a 20 µL reaction. Subsequent RT-PCR was performed using cDNA, Sybr Green PCR Master Mix (Life Technologies) and the following sense and antisense primers: *IL-1β* (5'-CAT GGA ATC CGT GTC TTC CT-3' and 5'-GAG CTG TCT GCT CAT TCA CG-3'), *IL-18* (5'-GAC CAA GGA AAT CGG CCT C-3' and 5'-GCC ATA CCT CTA GGC TGG CT-3') or *RPS26* (5'-AAG TTT GTC ATT CGG ATT AAC-3' and 5'-AGC TCT GAA TCG TGG TG-3') (0.5 pmol/µL). Real-time PCR was performed using the Applied Biosystems StepOne real-time PCR systems (Applied Biosystems) with the following profile: 10 min at 95°C, followed by a total of 40 two-temperature cycles (15 sec at 95°C and 1 min at 60°C). To verify the purity of the products, a melting curve was produced after each run according to the manufacturer's instructions. Results were expressed as fold induction after normalization by *RPS26*.

## In vivo light-challenge model

*Cx3cr1$^{GFP/GFP}$* mice were purchased from the Jackson Laboratories (Charles River Laboratories, Saint-Germain-sur-l'Arbresle, France). All mice were negative for gene defects for the *Crb1$^{rd8}$*, *Pde6b$^{rd1}$*, and *Gnat2$^{cpfl3}$* mutations. Mice were housed in the animal facility under specific pathogen-free condition, in a 12 hr/12 hr light/dark (100–500 lux) cycle with water and normal diet food available *ad libitum*. All experimental protocols and procedures were approved by the local animal care ethics committee (N°00156.02).

Two- to three-month-old mice were adapted to darkness for 6 hr, pupils dilated and exposed to green LED light (starting at 2AM, 4500 Lux, JP Vezon equipment, Yzeure, France) for 4 days as previously described (*Sennlaub et al., 2013*). Light-challenged mice were treated with daily subcutaneous injections of 100 µl of PBS or IL-1Ra (R&D; 1 mg/d/kg). On day 4, the mice were killed by CO2 asphyxiation and enucleated. The globes were fixed in 4% PFA for 30 min, then rinsed in 1x PBS (pH 7.3). Retinal and RPE/choroid tissues were dissected intact from the globe, stained with Peanut agglutinin Alexa 594 (Thermofisher; 1:50) and flat-mounted. Mononuclear phagocytes were visualized by using the Cx3cr1 promoter-controlled GFP expression.

## Quantification of cone outer segment volumes

Cone outer segment (OS) volumes were quantified using Imaris software (Bitplane) on stacks of confocal images which contain the entire photoreceptor outer and inner segments, of PNA-stained retinal flatmounts. 4 randomly chosen stacks (technical replicates) were taken per experimental retina or retinal explant. The software then calculated the volume of the PNA+ structures, which was subsequently divided by the number of PNA+ CS to yield the average individual CS volume in the stack. The CS volume of each retina or retinal explant (biological replicate) was calculated as the mean of the 4 individual stacks.

## Statistical analysis

Sample sizes for our experiments were determined according to our previous studies. Each experiment with mouse retinal explants or light-challenge retina is representative of at least 3 independent experiments. Graph Pad Prism 6 (GraphPad Software) was used for data analysis and graphic representation. Grubb's test was used to determine outliers. All values are reported as mean ± SEM. Statistical analysis was performed by one-way Anova analysis of variance followed by Bonferroni or

Dunnett's post-test (multiple comparison of groups with equal variance and normal distribution), Kruskal-Wallis followed by a Dunns post-test (non-parametric multiple comparison of groups) or Mann–Whitney $U$ test (non-parametric 2-group experiments) for comparison among means depending on the experimental design. The n and $p$-values are indicated in the figure legends.

## Acknowledgements

We would like to thank Stéphane Fouquet and David Godefroy from the plateforme d'imagerie of the Institut de la Vision (IdV) for invaluable assistance in confocal microscopy. We are grateful to and Céline Nouvel Jaillard (IdV) and to Claire Maëlle Fovet et Joanna Demilly (MIRCen platform, CEA/INSERM, Fontenay-aux-Roses, France) for organizing and giving us access to cynomolgus macaques post-mortem retina for retinal explants. This work was supported by grants from INSERM, ANR blanc (MACLEAR, ANR- J15R365), Labex Lifesenses, Carnot, and by Association de Prévoyance Santé de ALLIANZ. Dr. Craft is the inaugural endowed Mary D Allen Chair in Vision Research, Doheny Eye Institute, and supported, in part, by NIH EY015851, EY03040 (DEI), and Research to Prevent Blindness (USC Ophthalmology).

## Additional information

### Funding

| Funder | Grant reference number | Author |
|---|---|---|
| National Institutes of Health | EY015851 | Cheryl Mae Craft |
| National Institutes of Health | EY03040 | Cheryl Mae Craft |
| Association de Prévoyance Santé de ALLIANZ | | Florian Sennlaub |
| Agence Nationale de la Recherche | MACLEAR, ANR- J15R365 | Florian Sennlaub |
| Labex | LifeSenses | Florian Sennlaub |
| Carnot | International | Florian Sennlaub |

The funders had no role in study design, data collection and interpretation, or the decision to submit the work for publication.

### Author contributions

CME, XG, FS, Conception and design, Acquisition of data, Analysis and interpretation of data, Drafting or revising the article; HCM, Acquisition of data, Analysis and interpretation of data, Drafting or revising the article; SA, ED, SL, SJH, LS, Acquisition of data, Analysis and interpretation of data; VF, Acquisition of data; CMC, Contributed unpublished essential data or reagents; J-AS, RT, MP, Analysis and interpretation of data, Contributed unpublished essential data or reagents

### Author ORCIDs

Chiara M Eandi, http://orcid.org/0000-0003-3656-1689
Xavier Guillonneau, http://orcid.org/0000-0001-7379-3935
Florian Sennlaub, http://orcid.org/0000-0003-4412-1341

### Ethics

Human subjects: Volunteers provided written and informed consent for the human monocyte expression studies, which were approved by the Centre national d'ophthalmologie des Quinze-Vingt hospital (Paris, France) ethics committees (no. 913572)
Animal experimentation: All experimental protocols and procedures were approved by the local animal care ethics committee (00156.02; C 92-032-02 )

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
