## [Decision Letter]

Thank you for submitting your article "Subretinal mononuclear phagocytes induce cone segment loss via IL-1β" for consideration by *eLife*. Your article has been reviewed by three peer reviewers, and the evaluation has been overseen by Jeremy Nathans as the Reviewing Editor and a Senior Editor. The following individuals involved in review of your submission have agreed to reveal their identity: Joshua L Dunaief (Reviewer #1); Alfred Lewin (Reviewer #2).

Summary:

Thank you for submitting your manuscript "Subretinal mononuclear phagocytes induce cone segment loss via IL-1β" to *eLife*. The manuscript has been reviewed by three expert reviewers, and their assessments together with my own, forms the basis of this letter. I am including the three reviews at the end of this letter, as there are various specific comments in them that will not be repeated in the summary here.

All of the reviewers were impressed with the importance and novelty of your work. Overall, the experiments appear to be carefully executed. However, there is a general consensus that the data should be more cautiously interpreted and that various experiments that could strengthen (or weaken) a number of the conclusions are feasible.

We would like to encourage you to resubmit a revised manuscript that addresses the specific issues raised in the reviews.

Reviewer #1:

This paper focuses on the role of IL-1β secreted by infiltrating mononuclear cells in cone "segment" loss within the atrophy transition zone (TZ) from patients with dry age-related macular degeneration. Immunolabeling of post mortem eyes shows loss of cone segments in the TZ and association with subretinal D14+monocyte infiltration. Using human blood-derived CD14+monocytes in vitro and inflammation-prone *Cx3cr1^GFP/GFP^*mice in vivo, evidence is presented that monocyte derived IL-1β causes cone segment degeneration. Inhibiting subretinal monocyte accumulation or Il-1β might protect the cone segments and help preserve or restore high acuity vision in AMD.

The paper is conceptually and translationally important, but would benefit from addressing several limitations.

1) In Figure 1, cone arrestin is used to visualize cone "segments." Yet, arrestin translocation with light exposure and redistribution after retinal detachment suggests that arrestin-poor segments might still be present. It would be helpful to also show a phase-contrast image.

2) The rationale for use of both arrestin (to label cone segments and PNA to label cone segment sheaths in Figure 2 and beyond should be stated explicitly. The use of PNA in Figure 1 could also help with the issues raised in point #1.

3) Does cone "segments" refer to outer segments, inner segments, or both?

4) The authors "carefully dissected" Bruch's membrane from the photoreceptors in samples from geographic atrophy eyes prior to immunostaining/imaging retinal flat mounts. Was this dissection limited to the atrophic zone and not the nearby transition zone (within 1-200 microns of the atrophic zone)? Such dissection has the potential to strip outer segments from the imaged tissue.

Reviewer #2:

This paper by Sennlaub and colleagues builds on their earlier work showing that blood derived mononuclear cells contribute to the pathology of age related macular degeneration (AMD) and perhaps also to the death of cone photoreceptors in retinitis pigmentosa (RP). This work is of great clinical significance since AMD is the leading cause of blindness in older adults, at least in the developed world, and RP is the leading inherited form of blindness. What is new in this paper is the demonstration that the loss of cone photoreceptor inner and outer segments in the transition zone coincides with the infiltration of CD14+ monocytes in donor eyes from patients with advanced dry AMD. Looking for a mechanism, they turned to retinal explants from Cx3cr1 mutant mice and showed that CD14+ monocytes led to rod cell death (TUNEL positive nuclei) and degradation of cone outer and inner segments. They went on to show that this damage may be caused by the release of IL-1beta from the monocytes.

The histology in the paper is excellent and their efforts to present the data in a quantitative fashion were also good.

The authors should address a few issues, however:

1) It seemed a bit odd to use human monocytes on the mouse explants. In previous work focused on the impact on rod cells, they used mouse monocytic cells. Why not use mouse on mouse for this study? In their earlier work did they observe an impact on cone outer or inner segments?

2) They used 50 ng/ml of recombinant IL-1beta to damage cone photoreceptors in explants, but they never reported the level of this cytokine produced by the monocytes in their co-culture conditions. They should do this and use an equivalent amount of the recombinant protein.

3) How does the cone density and the ratio of cones to rods differ between the mouse explants and the atrophic and transitional zones studied in human samples?

Reviewer #2 (Additional data files and statistical comments):

Their statistical analysis was appropriate.

Reviewer #3:

Eandi et al. propose that subretinal monocytic phagocytes (MPs) that accumulate in the junctional zone of geographic atrophy secrete IL1b which causes cone segment loss. This group, which has been a leader in forwarding the role of subretinal MPs in AMD, previously showed that MPs contribute to rod loss, and that a subset of MPs are blood derived Monocytes. The concept is intriguing, and if correct, would shed new insights into vision loss in GA due to cone segment loss. A number of methodologic and conceptual concerns are raised by this reviewer.

The main novel information is that IL1b injures cone segments in the JZ of GA. In its present form, this reviewer does not think that the authors have provided enough data to support this conclusion. To more convincingly prove this point, the authors should provide data identifying the topographical location of IL1b in their human GA samples. The IL1b assessment would be significantly strengthened with a more in depth analysis (see below).

In Figure 1, the variable distance of the TZ could bias the morphometric semi-quantification since there is likely a gradient of morphometric severity closer to the AZ. The analysis would be strengthened by standardizing a specific distance from the AZ where measurments are made.

In Figure 2, some methodological concerns are raised: 1) a larger area of the flatmount should be shown since flatmounts can have very heterogeneous distribution of changes; 2) semi-quantification of the number of MPs would greatly enhance the findings for controls, >1000um, TZ, and AZ areas that would support the conclusions; 3) the choice of CD14 labeling makes the interpretation of MPs challenging because the RPE expresses CD14 (Elner et al. Exp Eye Res, 2003). How do the authors know that the labeling is solely attributed to MPs? Confirming with another monocyte marker appears necessary. 4) How do the authors separate MCs from subretinal MPs, especially since activated MCs are more truncated? This is an important conclusion of the manuscript, and thus, has not been proven. 5) In the images, there does not appear to be much difference in the number of MPs in the TZ from AZ. 6) in 2C and 2D, why is there such a paucity of arrestin stained cone cells in the controls?

While Figure 3 illustrates that MPs can induce rod apoptosis and cone changes, using 100,000 human macrophages does not seem physiologically relevant. What happens when fewer cells are used, perhaps in the range of what is seen in GA? Why do the authors use human MPs? Wouldn't this be expected to elicit a reaction due to different species? What happens when mouse MPs are used? Why don't the authors use double labeling of PNA and arrestin throughout the figures?

Quantifying IL-1b mRNA is not sufficient for determining its ultimate impact since it is regulated not only transcriptionally, as the authors have demonstrated, but also post-translationally by the inflammasome. Thus, the authors should provide evidence that the MPs secrete mature IL1b protein. To understand the full impact of IL-1b, further analysis is necessary: 1) It is the ratio of IL-1b/IL18 that is highly relevant because acutely, as in these experiments, IL18 can neutralize some of the pro-inflammatory effects of IL1b. Thus, the authors should demonstrate that IL18 does not change. 2) a dose response effect of IL1b would be interesting to determine any threshold effect, especially since 50 ng/ml is strikingly supraphysiologic. What happens if the IL1b is in a more physiologic range? While not absolutely necessary, providing evidence of inflammasome activation would be a very nice compliment that would drive home this point.

Figure 5 provides evidence that IL1b is involved in PNA positive shortening through IL1R antagonist treatment. It is important to describe how the area studied was determined since, as mentioned above, there can be tremendous heterogeneity on flatmounts. The authors should provide quantitative data showing no change in the number of monocytes with IL1b inhibition. In their prior work, they showed this effect in their chronic model, and here they only show their acute light tox model. This is a short-coming since they are attempting to extrapolate these data to a chronic disease.

While the authors mention the possibility of an RPE contribution, they do not provide any studies to support this notion. Thus, in the Discussion, they should more fully acknowledge that they can't rule out RPE cell dysfunction, especially since the RPE is clearly dysmorphic in their JZ GA samples, as a source of rod and cone loss and damage (i.e. Figure 1).

---

## [Author Response]

*Reviewer #1:*

*The paper is conceptually and translationally important, but would benefit from addressing several limitations.*

1) In Figure 1, cone arrestin is used to visualize cone "segments." Yet, arrestin translocation with light exposure and redistribution after retinal detachment suggests that arrestin-poor segments might still be present. It would be helpful to also show a phase-contrast image.

In mice, a light-dependent translocation of cone arrestin has been demonstrated, and we agree with the reviewer that arrestin-poor segments might give the false impression of cone segment loss in arrestin immunohistochemistry. As Bird et al. (2014) already demonstrated cone segment loss in the TZ by classical histology, we here performed L/M cone opsin immunohistochemistry to verify CS loss using a second marker. These additional immunohistochemistries confirm the loss of CS and are presented as additional insets in Figure 1. The text in the Results section was altered accordingly: “L/M-cone opsin immunohistochemistry (Figure 1 insets), which recognizes the opsins of the most abundant red and green cones, confirmed the CS loss in the TZ and AZ.”

2) The rationale for use of both arrestin (to label cone segments and PNA to label cone segment sheaths in Figure 2 and beyond should be stated explicitly. The use of PNA in Figure 1 could also help with the issues raised in point #1.

We introduced a sentence in the Results section that explains the meaning of the different antibodies, which also relates to the point 3 raised by the reviewer: “We used CD14 immunohistochemistry to visualize MPs, peanut agglutinin (PNA) that stains inner and outer cone segments (but not cone cell bodies), and cone arrestin that stains cones irrespective of the presence of CS (see Figure 1).”

3) Does cone "segments" refer to outer segments, inner segments, or both?

PNA, cone opsins, or cone arrestin (Figure 1) all mark both, the inner and outer cone segments. The differentiation of the two in healthy retina can be made morphologically as the inner and outer segment are separated by the cilium (Blanks and Johnson, 1984). In the AZ and TZ of GA patients, where the cellular architecture is disturbed it is often not possible to distinguish a remaining inner segment from the rest of the cone cell body. It is therefore difficult to say to what extend the inner segments remain, which is why we generally referred to cone segments. In the revised version we changed the corresponding paragraph to make this difficulty more clear:

“In the TZ, close to the margin of the lesion but where RPE was still present, the ONL was irregular, thinned (around 2 nuclei of ONL), and cone and rod morphology was severely altered with the outer segments missing and the inner segments difficult to distinguish (Figure 1).”

4) The authors "carefully dissected" Bruch's membrane from the photoreceptors in samples from geographic atrophy eyes prior to immunostaining/imaging retinal flat mounts. Was this dissection limited to the atrophic zone and not the nearby transition zone (within 1-200 microns of the atrophic zone)? Such dissection has the potential to strip outer segments from the imaged tissue.

In GA tissues, the retinas of the AZ adhere to Bruchs membrane in the area of RPE defects and needed to be carefully peeled for separation. This is not the case for the TZ, where the retina detaches easily from the underlying RPE, comparable to control retinas. We agree with the reviewer that dissecting the tissues always holds a risk of altering the morphology. However, the loss of CS is also observed in the paraffin sections (Figure 1) and has previously been described in histology and electron microscopy, which strongly suggests that CS loss is not an artifact (Bird et al., 2014). We changed the text in the wording in the Materials and methods section to make this more clear:

“In GA tissues, the retinas of the AZ were carefully peeled from the RPE/choroid, as they adhere to Bruchs membrane in the area of RPE defects. This is not the case for the TZ, where the retina detaches easily from the underlying RPE, comparable to control retinas.”

*Reviewer #2:*

*1) It seemed a bit odd to use human monocytes on the mouse explants. In previous work focused on the impact on rod cells, they used mouse monocytic cells. Why not use mouse on mouse for this study?*

We thought it would be particularly interesting to use human Mo but we agree with the reviewer that interspecies incompatibilities could lead to artifacts, even though a xenograft rejection type of mechanism requires antibodies (hyperacute reaction) and/or NKs, T-lymphocytes of the adaptive immune system (chronic rejection), not present in our 18h co-culture system.

To evaluate eventual interspecies incompatibilities we performed two sets of experiments: 1.) we quantified cone segments in cocultures of Cx3cr1^GFP/GFP^BMMs with mouse retinas and 2.) we cultivated human blood derived Mos with non-human primate (macaque) retinal explants.

These new results are presented in the new Figure 3.

*In their earlier work did they observe an impact on cone outer or inner segments?*

Maybe surprisingly, we did very little PNA stains in Cx3cr1-deficient animals. I remember doing some PNA flatmount stains back in 2007. As we did not notice any differences in cone numbers; we hastily concluded that there was no effect on cones at the time.

*2) They used 50 ng/ml of recombinant IL-1beta to damage cone photoreceptors in explants, but they never reported the level of this cytokine produced by the monocytes in their co-culture conditions. They should do this and use an equivalent amount of the recombinant protein.*

In these experiment we used IL-1β at a concentration of 50ng/ml, which is in the range of concentration classically used in vitro. The local IL-1β concentration that cones are exposed to in Mo/retina co-cultures is difficult to estimate, as a gradient of IL-1β is expected to form around the IL-1β producing Mos. This is the reason why we directly evaluated the effect of locally produced IL-1β using IL-1 receptor antagonist (IL-1Ra) on the co-cultures.

We introduced two explanatory sentences in the corresponding Results section to better explain this point: “In these experiment IL-1β was used at a concentration of 50ng/ml (in the range of concentration classically used in vitro) but the local IL-1β concentration that cones are exposed to in the co-culture is difficult to estimate, as a gradient of IL-1β is expected to form around the IL-1β producing Mos. To directly evaluate the effect of locally produced IL-1β we evaluated the effect of the IL-1 receptor antagonist (IL-1Ra) on the co-cultures.”

3) How does the cone density and the ratio of cones to rods differ between the mouse explants and the atrophic and transitional zones studied in human samples?

We agree with the reviewer that the co-culture in vitro system is (as most in vitro models) only a rough approximation of the human situation. It is characterized by an excess of Mos, the lack of underlying RPE, mild rod apoptosis and quite severe CS degeneration occurring in hours rather than days and weeks. However, the pattern of continuous rod (but not cone) apoptosis and CS degeneration over longer time periods would likely result in severe rod cell loss (as they don’t regenerate), little cone loss but CS degeneration intriguingly similar to what is observed in the TZ of GA patients. To make this differences and similarities more clear to the reader, we introduced an additional paragraph in the Discussion:

“The co-culture in vitro system, characterized by an excess of Mos, the lack of underlying RPE, mild rod apoptosis and quite severe CS degeneration occurring in hours rather than days and weeks, can however only be a rough approximation of the in vivo situation. However, the pattern of continuous rod (but not cone) apoptosis and CS degeneration over longer time periods would result in severe rod cell loss (as they don’t regenerate), little cone loss but CS degeneration intriguingly similar to what is observed in the TZ of GA patients.”

*Reviewer #3:*

*Eandi et al. propose that subretinal monocytic phagocytes (MPs) that accumulate in the junctional zone of geographic atrophy secrete IL1b which causes cone segment loss. This group, which has been a leader in forwarding the role of subretinal MPs in AMD, previously showed that MPs contribute to rod loss, and that a subset of MPs are blood derived Monocytes. The concept is intriguing, and if correct, would shed new insights into vision loss in GA due to cone segment loss. A number of methodologic and conceptual concerns are raised by this reviewer.*

The main novel information is that IL1b injures cone segments in the JZ of GA. In its present form, this reviewer does not think that the authors have provided enough data to support this conclusion. To more convincingly prove this point, the authors should provide data identifying the topographical location of IL1b in their human GA samples. The IL1b assessment would be significantly strengthened with a more in depth analysis (see below).

We agree with the reviewer that our data shows an association of subretinal MPs with CS degeneration in the TZ of AMD and that MPs can induce a similar phenotype in vitro and in vivo, which strongly suggest that subretinal MPs participate in the degenerative changes in GA, but not necessarily that this effect is mediated by IL-1β in GA. Our attempts to detect IL-1β in our human tissue samples, including tonsillitis preparations that served as positive controls, have so far failed to establish a reproducible protocol to study its expression in GA. However, IL-1β has been shown to be strongly expressed in subretinal MPs of choroidal neovascular membranes in wet AMD (Oh et al., 1999) and CS degeneration is also observed adjacent to the fibrovascular membranes in disciform subretinal scars of wet AMD(Curcio et al., 2001). These observations might suggest that an IL-1β-dependent mechanism might exist in the human pathology. We introduced a paragraph in the Discussion to make this point very clear to the reader:

“Our study clearly demonstrates the association of subretinal MPs with CS degeneration in human samples and that MPs can induce a similar phenotype in vitro and in vivo, which strongly suggest that subretinal MPs participate in the degenerative changes in GA. Further studies of IL-1β expression in human GA samples are needed to implicate this specific cytokine in the degenerative process in the human disease. However, IL-1β has been shown to be strongly expressed in subretinal MPs of choroidal neovascular membranes in wet AMD and CS degeneration is also observed adjacent to the fibrovascular membranes in disciform subretinal scars of wet AMD. These observations might suggest that an IL-1β-dependent mechanism might exist in the human pathology.”

In Figure 1, the variable distance of the TZ could bias the morphometric semi-quantification since there is likely a gradient of morphometric severity closer to the AZ. The analysis would be strengthened by standardizing a specific distance from the AZ where measurments are made.

We agree with the reviewer that the TZ is of variable length (200-800μm) in the human sections. This has been previously reported (Bird et al., 2014). In our quantification we actually did exactly what the reviewer is suggesting. We counted photoreceptors in the AZ, in the 200μm of the TZ adjacent to the AZ as all TZs were at least 200μm wide on the sections, and at >1000μm of the AZ, as none of the TZs in our samples exceeded 800μm. This is described in the corresponding Results section:

“The TZ, characterized by a thinned ONL in the presence of underlying RPE, was of variable length (200-800μm) in our samples similar to previous reports. […]The cone density at a greater than 1000μm distance from the RPE-denuded AZ, was similar between control and GA eyes. In the atrophic zone the number of cone somata was significantly reduced to around half the numbers of controls (Figure 1) and was not significantly different in the TZ (0-200μm from the AZ, which is the TZ length invariably found in all donor eyes).”

*In Figure 2, some methodological concerns are raised: 1) a larger area of the flatmount should be shown since flatmounts can have very heterogeneous distribution of changes; 2) semi-quantification of the number of MPs would greatly enhance the findings for controls, >1000um, TZ, and AZ areas that would support the conclusions;*

We agree with the reviewer and mention in the manuscript that the TZ is of variable size (see above). We have previously quantified the number of subretinal IBA1^+^ and CCR2^+^MPs on the paraffin sections of the AZ of the GA samples and controls and illustrated their presence in the TZ on paraffin sections (Sennlaub et al., 2013) and flatmounts (IBA1 stain) (Levy et al., 2015)and this is cited in the manuscript. In Figure 2 we describe an association of subretinal MPs with morphological changes in the TZ. We believe that our previous reports of MPs in the TZ, the morphological description of photoreceptor changes in the paraffin sections (Figure 1) and the flatmount immunohistochemistry of both, MPs and photoreceptor changes, clearly demonstrate that subretinal MPs are associated with photoreceptor morphological changes in the TZ.

Showing larger areas of the flatmounts in the figures is problematic as one cannot discern the CD14+cells or CS anymore. We therefore tried to find a compromise by showing a larger area for the RPE flatmounts to illustrate the infiltration of the TZ by MPs and closer views of the subretinal aspects of the retinal flatmounts to illustrate the morphological changes in photoreceptors and their close relation to subretinal MPs.

*3) the choice of CD14 labeling makes the interpretation of MPs challenging because the RPE expresses CD14 (Elner et al. Exp Eye Res, 2003). How do the authors know that the labeling is solely attributed to MPs? Confirming with another monocyte marker appears necessary.*

We agree with the reviewer that the RPE also expresses CD14. However the expression level in RPE is greatly inferior compared to Mos (which is shown in Figure 4 of Elner et al. (2003) and which our RT-PCR of mRNA from circulating and cultured human monocytes and post-mortem human RPE samples confirm). This difference likely explains why the RPE did not appear CD14-positive in our experimental conditions. We added higher magnification of orthogonal and perpendicular 3D reconstruction pictures of confocal Z-stack micrographs to Figure 2 that illustrates the morphology and positioning of the subretinal CD14+cell on the apical side of the RPE that does not appear CD14-positive in our flatmount immunohistochemistry. Additionally, we provide CD14/IBA-1 double labeling on retinal flatmounts (Figure 2), to demonstrate that CD14+ cells are also positive for other MP-markers. The description of these additional results were added to the corresponding Results section.

*4) How do the authors separate MCs from subretinal MPs, especially since activated MCs are more truncated? This is an important conclusion of the manuscript, and thus, has not been proven.*

We have previously reported that the population of subretinal MPs is in part derived from infiltrating CCR2^+^blood monocytes (Sennlaub et al., 2013) and likely in part from resident MPs. We do not claim that CD14 specifically recognizes infiltrating MPs, evidenced by our results of CD14+MCs in the control retina (2B). We carefully rephrased any sentence that might have given the wrong impression that this was our intent throughout the revised manuscript.

*5) In the images, there does not appear to be much difference in the number of MPs in the TZ from AZ.*

In the flatmount preparations, some of subretinal MP population will stick to the RPE flatmount, while others stick to the retinal side, as shown in Figure 2. This might depend on individual sample constitution, fixation and other incontrollable factors. We have previously quantified the numbers of MPs in the AZ (Sennlaub et al., 2013) and they seem to be generally more numereous in the AZ than in the TZ. In this manuscript we describe an association of subretinal MPs with morphological changes in the TZ. We believe that our previous reports of MPs in the TZ, the morphological description of photoreceptor changes in the paraffin sections (Figure 1) and the flatmount immunohistochemistry of both, MPs and photoreceptor changes, clearly demonstrate that subretinal MPs are associated with photoreceptor morphological changes in the TZ.

Author response image 1.Additional examples of CD14-stained RPE flatmounts**DOI:**
http://dx.doi.org/10.7554/eLife.16490.007

*6) in 2C and 2D, why is there such a paucity of arrestin stained cone cells in the controls?*

All PNA^+^CS also stain positive for cone arrestin. To illustrate this we introduced PNA and arrestin staining separately as insets of the new Figure 2. In both the AZ and TZ the typical cone arrestin^+^PNA^+^cone segment pattern had disappeared and cone arrestin immunological staining pattern of the remaining cone somata became apparent in the flat-mount, possibly because of greater antibody penetration in the thinned retinal flatmount.

*While Figure 3 illustrates that MPs can induce rod apoptosis and cone changes, using 100,000 human macrophages does not seem physiologically relevant. What happens when fewer cells are used, perhaps in the range of what is seen in GA?*

We performed additional experiments evaluating the effect of different numbers of Mos on CS, that are presented in the new Figure 3. Please note and we rephrased this in the new manuscript that the co-culture is an in vitro model. We are fully aware that the co-culture in vitro system, is characterized by an excess of Mos. This is the reason why we performed the in vivo experiments, to verify if these changes can be observed with naturally occurring subretinal inflammation and an underlying RPE cell layer.

*Why do the authors use human MPs? Wouldn't this be expected to elicit a reaction due to different species? What happens when mouse MPs are used?*

We thought it would be particularly interesting to use human Mo but we agree with the reviewer that interspecies incompatibilities could lead to artifacts, even though a xenograft rejection type of mechanism requires antibodies (hyperacute reaction) and/or NKs, T-lymphocytes of the adaptive immune system (chronic rejection), not present in our 18h co-culture system.

To evaluate eventual interspecies incompatibilities we performed two sets of experiments: 1.) we quantified cone segments in cocultures of Cx3cr1^GFP/GFP^BMMs with mouse retinas and 2.) we cultivated human blood derived Mos with non-human primate (macaque) retinal explants.

These new results are presented in the new Figure 3.

Why don't the authors use double labeling of PNA and arrestin throughout the figures?

The new Figure 3 contains detailed views of the arrestin staining that confirms the CS degeneration.

*Quantifying IL-1b mRNA is not sufficient for determining its ultimate impact since it is regulated not only transcriptionally, as the authors have demonstrated, but also post-translationally by the inflammasome. Thus, the authors should provide evidence that the MPs secrete mature IL1b protein.*

The IL-1beta (IL-1β) gene is transcribed after an inflammatory stimulus (such as toll like receptor {TLR2/4} stimulation) and synthetized as the IL-1β precursor. Activation by a second stimulus, such as the P2RX7 receptor by extracellular adenosine triphosphate (ATP), can trigger the assembly of the NLRP3 inflammasome, and activate caspase-1 that cleaves IL-1β precursor and produces mature, secretable IL-1β (Schroder and Tschopp, 2010). Human blood monocytes, that we used in our experiments, constitutively release endogenous ATP, which activates caspase-1 and leads to mature IL-1β after transcriptional induction without a second exogenous stimulus (Netea et al., 2009). Similarly, we showed that mouse bone marrow-derived monocytes (BMMs) from Cx3cr1-deficient mice, contrary to wildtype BMMs, also constitutively secrete ATP, express increased surface P2RX7 receptor and secrete mature IL-1β after transcriptional stimulation without a second stimulus (Hu et al., 2015). The local IL-1β concentration that cones are exposed to in Mo/retina co-cultures is difficult to estimate, as a gradient of IL-1β is expected to form around the IL-1β producing Mos. This is the reason why we directly evaluated the effect of locally produced IL-1β using IL-1 receptor antagonist (IL-1Ra) on the co-cultures.

In vivo, we previously demonstrated that IL-1β protein is increased after the 4 day acute light-challenge in *Cx3cr1^GFP/GFP^*mice.

We introduced two explanatory sentences in the corresponding Results section to better explain this point: “In these experiment IL-1β was used at a concentration of 50ng/ml (in the range of concentration classically used in vitro) but the local IL-1β concentration that cones are exposed to in the co-culture is difficult to estimate, as a gradient of IL-1β is expected to form around the IL-1β producing Mos. To directly evaluate the effect of locally produced IL-1β we evaluated the effect of the IL-1 receptor antagonist (IL-1Ra) on the co-cultures.”

*To understand the full impact of IL-1b, further analysis is necessary: 1) It is the ratio of IL-1b/IL18 that is highly relevant because acutely, as in these experiments, IL18 can neutralize some of the pro-inflammatory effects of IL1b. Thus, the authors should demonstrate that IL18 does not change.*

The new Figure 4 shows that IL18 is not induced in hMo in the co-culture.

2) a dose response effect of IL1b would be interesting to determine any threshold effect, especially since 50 ng/ml is strikingly supraphysiologic. What happens if the IL1b is in a more physiologic range? While not absolutely necessary, providing evidence of inflammasome activation would be a very nice compliment that would drive home this point.

In this experiment we used IL-1β at a concentration of 50ng/ml, which is in the range of concentration classically used in vitro. The local IL-1β concentration that cones are exposed to in Mo/retina co-cultures is difficult to estimate, as a gradient of IL-1β is expected to form around the IL-1β producing Mos. This is the reason why we directly evaluated the effect of locally produced IL-1β using IL-1 receptor antagonist (IL-1Ra) on the co-cultures and the experiments in vivo.

Concerning the inflammasome, and as mentioned above, human blood monocytes, that we used in our experiments, constitutively release endogenous ATP, which activates caspase-1 and leads to mature IL-1β after transcriptional induction without a second exogenous stimulus (Netea et al., 2009). Similarly, we showed that mouse bone marrow-derived monocytes (BMMs) from Cx3cr1-deficient mice, contrary to wildtype BMMs, also constitutively secrete ATP, express increased surface P2RX7 receptor and secrete mature IL-1β after transcriptional stimulation without a second stimulus (Hu et al., 2015).

*Figure 5 provides evidence that IL1b is involved in PNA positive shortening through IL1R antagonist treatment. It is important to describe how the area studied was determined since, as mentioned above, there can be tremendous heterogeneity on flatmounts.*

We explain in the Materials and methods section how this quantifications are performed: “Cone outer segment (OS) volumes were quantified using Imaris software (Bitplane) on stacks of confocal images which contain the entire photoreceptor outer and inner segments, of PNA-stained retinal flatmounts. 4 randomly chosen stacks (technical replicates) were taken per experimental retina or retinal explant.”

The authors should provide quantitative data showing no change in the number of monocytes with IL1b inhibition. In their prior work, they showed this effect in their chronic model, and here they only show their acute light tox model. This is a short-coming since they are attempting to extrapolate these data to a chronic disease.

The new Figure 5 represents the subretinal MP quantification in the control and light-challenged wildtype- and Cx3cr1^GFP/GFP^-mice with and without the IL-1RA treatment. We chose to use the light-challenge model as daily injections of subcutaneous IL-1RA injections over a 12 months time period were too difficult to perform.

*While the authors mention the possibility of an RPE contribution, they do not provide any studies to support this notion. Thus, in the Discussion, they should more fully acknowledge that they can't rule out RPE cell dysfunction, especially since the RPE is clearly dysmorphic in their JZ GA samples, as a source of rod and cone loss and damage (i.e. Figure 1).*

We fully agree with the reviewer that the photoreceptor degeneration in AMD is likely the result of several factors and acknowledge this in the manuscript: “It is not clear why CS degenerate in the TZ, where the RPE is present. The degeneration might be a result of dysfunctional RPE prior to its degeneration, suggested by the observation of hyperfluorescence of the TZ. […] The observation that CS degeneration is also observed in RP patients with rod-gene mutations and unremarkable RPE further argues that a RPE-independent factor might participate in the degeneration.”